# Association of Vitamin C Administration with Postoperative Delirium After Cardiac Surgery with Cardiopulmonary Bypass: A Single-Center Retrospective Exploratory Cohort Study [note 1]

**DOI:** 10.3390/jcm15010135

**Published:** 2025-12-24

**Authors:** Yoshihide Kuribayashi, Shigekiyo Matsumoto, Yoshifumi Ohchi, Shinya Kai, Yoshimasa Oyama, Tetsuya Uchino, Osamu Tokumaru, Chihiro Shingu

**Affiliations:** 1Department of Anesthesiology, Faculty of Medicine, Oita University, Yufu 879-5593, Japan; kuribayashi@oita-u.ac.jp (Y.K.); oyama@oita-u.ac.jp (Y.O.);; 2Department of Physiology, Faculty of Welfare and Health Sciences, Oita University, Oita 870-1192, Japan; ostokuma@oita-u.ac.jp

**Keywords:** postoperative delirium, vitamin C, cardiac surgery, cardiopulmonary bypass, electron spin resonance, dimethyl sulfoxide, oxidative stress, reactive oxygen species, inflammation

## Abstract

**Objectives:** Oxidative stress after cardiac surgery may disrupt the blood–brain barrier and contribute to postoperative delirium (POD). Although associations between oxidative stress and POD are recognized, whether vitamin C (VC) can prevent POD remains poorly understood. This study aimed to explore the association of VC administration with POD after cardiac surgery. **Methods:** Eighty-four patients undergoing elective cardiac surgery at our hospital were enrolled. The non-VC group (NVC, n = 40) consisted of patients treated between October 2021 and March 2022, while the VC group (n = 44) included those treated between April and September 2022 who received 2 g intravenous VC at intensive care unit (ICU) admission. The primary outcome was POD incidence. Electron spin resonance (ESR) measured AFR/DMSO, which reflected VC before induction, after CPB withdrawal, at ICU admission, and on postoperative day 1. **Results:** Baseline characteristics, comorbidities, and intraoperative factors were similar between groups. Postoperative organ dysfunction and inflammation were also comparable, although lactate levels were 40% higher in the VC group. POD incidence was significantly lower with VC (35.0% vs. 11.4%, *p* < 0.01). Logistic regression analysis confirmed that VC reduced POD risk (adjusted odds ratio 0.22, 95% CI 0.07–0.69, *p* < 0.01). ESR showed that postoperative AFR/DMSO levels dropped sharply but normalized by day 1 in VC-treated patients. **Conclusions:** This study suggests that 2 g of VC administered at ICU admission may reduce POD incidence. In the future, these findings require confirmation in randomized trials.

## 1. Introduction

Postoperative delirium (POD) is an acute neurocognitive disorder characterized by impaired consciousness and cognitive decline, and it is a common complication in older patients. As the population ages, POD incidence is expected to rise. Incidence rates following cardiac surgery are particularly high, reaching up to 52% [1]. Moreover, delirium after cardiac surgery is a risk factor for developing dementia within 5 years [2]. Because POD is also associated with increased mortality and prolonged hospitalization [3], establishing preventive methods is a pressing issue for improving the prognosis in cardiac surgery patients.

No conclusive pharmacotherapy for POD exists because its pathogenic mechanism remains unclear. However, animal experiments suggest the involvement of both neuroinflammation and oxidative stress [4,5,6]. In cardiac surgery using cardiopulmonary bypass (CPB) with cardiac arrest, systemic inflammation, oxidative stress, and apoptosis occur following ischemia–reperfusion injury and neutrophil activation, impairing remote organ function and prognosis [7]. Ischemia–reperfusion injury arises when reactive oxygen species (ROS)—produced by enzymes such as xanthine oxidase and NADPH oxidase—damage mitochondrial function. Excessive cytokine release during systemic inflammatory response further activates neutrophils and promotes the expression of adhesion molecules on vascular endothelium. After adhering to the vascular endothelium, activated neutrophils release ROS and proteases that damage the endothelium, leading to microangiopathy and resulting in dysfunction of both the heart and remote organs. Through activation of NF-κB and MAPK signaling pathways, ROS also induce apoptosis and inflammation, thereby indirectly promoting cell injury [7]. Recent reports have shown that oxidative stress after cardiac surgery is associated with POD [8,9]. Therefore, controlling oxidative stress is important for preventing POD.

Dexmedetomidine and melatonin, which have shown efficacy in POD, possess antioxidant properties; however, antioxidant activity is not their primary mechanism of action. Vitamin C (VC) is a well-established antioxidant and oxidative stress marker. Of all antioxidants in the body, it reacts most rapidly with ROS and is depleted in a short period of time [10]. Because VC cannot be synthesized endogenously [11], adequate perioperative supplementation requires intravenous administration. We selected VC for this study because it is an endogenous antioxidant whose safety in cardiac surgery has been supported by several studies [12]. However, the dosage, timing, and duration of VC administration during cardiac surgery have been rarely investigated [12]. Maximum oxidative stress typically occurs immediately after CPB, as reflected by increases in malondialdehyde [13]. However, this time point often coincides with uncontrolled bleeding and unstable hemodynamics. Therefore, we considered postoperative intensive care unit (ICU) admission—when bleeding is controlled and hemodynamics are stabilized—the optimal time for VC administration. Although dosing regimens in prior cardiac surgery studies have been highly heterogeneous, many randomized controlled trials (RCTs) have demonstrated that approximately 2 g per day can be administered safely without significant adverse effects [12]. Based on this evidence, we hypothesized that intravenous administration of 2 g VC at ICU admission would reduce POD incidence by restoring antioxidant capacity and attenuating oxidative stress.

However, ROS generated by NADPH oxidase in macrophages not only play an essential role in host defense but also contribute to intracellular signal transduction [14]. Oversuppression of ROS is therefore undesirable [15]. We previously developed a method to measure ascorbyl free radical/dimethyl sulfoxide (AFR/DMSO), which reflects VC levels in real-time, using electron spin resonance spectroscopy (ESR) [16]. This method helps prevent overdosing on VC.

In the present study, we used ESR to measure VC in real time [16] and conducted a retrospective before-and-after study to examine the inhibitory effect of intravenous VC (2 g) administered at postoperative ICU admission on POD in patients undergoing cardiac surgery with CPB. This study aimed to test whether a single 2 g intravenous dose of VC administered at ICU admission reduces the incidence of POD in patients undergoing cardiac surgery with CPB.

## 2. Materials and Methods

### 2.1. Study Design and Setting

This single-center, retrospective, time-based cohort study compared two independent patient groups—before and after the initiation of VC administration upon ICU admission—rather than repeated measurements in the same patients. The study was approved by the Oita University Hospital Ethical Review Board (Approval No. 2307) and conducted in accordance with the Declaration of Helsinki. In line with the instructions of the Ethical Review Board, patient informed consent was waived, and the study details were published on the Department of Anesthesiology’s website.

The study population included patients admitted to the ICU at Oita University Hospital after undergoing elective CPB-assisted cardiac surgery. Patients younger than 18 years and pregnant patients were excluded. Based on the start of VC administration, patients were divided into two groups: the non-VC group (NVC, n = 40), comprising patients between 1 October 2021, and 31 March 2022, and the VC group (VC, n = 44), comprising patients between 1 April 2022, and 30 September 2022, who received VC upon postoperative ICU admission (Figure 1). Grouping was based on the calendar time. No significant changes were observed in ICU care protocols, staffing, or delirium assessment procedures across the study periods, according to ICU medical records. Nevertheless, the possibility of temporal bias due to unmeasured seasonal variations or other practice changes cannot be entirely excluded and should be considered a limitation.

All blood test and blood gas analysis data were obtained through routine postoperative testing. For ESR-based VC measurements, the blood sample left in the syringe after routine blood gas analyses was used in both groups. Plasma remaining after measurement was frozen at −80 °C. No additional blood samples were collected for AFR/DMSO or cytokine measurements.

### 2.2. Intervention

At our ICU, to reduce excessive oxidative stress postoperatively, we added a treatment protocol with the antioxidant VC administration upon ICU admission following cardiac surgery under CPB starting from April 2022. This study was designed in alignment with these changes and retrospectively compared the incidence of POD before and after initiating VC administration.

VC was administered as an ascorbic acid injection (Vitacimin^®^, Takeda Pharmaceutical Co., Ltd., Osaka, Japan). A single dose of 2 g was administered via a central venous catheter as an intravenous bolus at ICU admission after surgery (Figure 2). Administration was performed by ICU physicians and recorded in the electronic health record. No VC was administered intraoperatively.

### 2.3. Anesthesia and ICU Management Protocol

Before induction, a radial arterial catheter was inserted for invasive blood pressure monitoring and blood sampling. In both groups, anesthesia was induced with propofol, analgesia was provided with fentanyl or remifentanil, and muscle relaxation was achieved with rocuronium. Anesthesia was maintained with total intravenous administration of remifentanil and propofol. Catecholamines and vasodilators were administered as needed to maintain mean arterial pressure ≥ 60 mmHg. Ventilator settings targeted SpO_2_ of 94–99% and EtCO_2_ of 35–45 mmHg. Depth of anesthesia was adjusted to maintain BIS between 40 and 60. After surgery, all patients were admitted to the ICU. Selection of sedatives, including propofol, midazolam, or dexmedetomidine, was at the discretion of the intensivist. Patients were extubated once hemodynamic stability was achieved, and SpO_2_ was maintained at 94–99% with supplemental oxygen post-extubation.

### 2.4. Outcome Definitions and Assessment Protocols

The primary outcome was the incidence of POD during ICU admission. POD assessment began immediately upon ICU admission and continued throughout the ICU stay. POD was defined as a positive CAM-ICU [17] or an ICDSC score ≥ 4 [18]. Delirium assessments were performed at least four times daily by ICU nurses trained in CAM-ICU and ICDSC evaluation, and positive results were confirmed by intensivists. Patients with even a single positive CAM-ICU or ICDSC assessment were considered to have developed POD. Interrater reliability was ensured through standardized training, and ICU nurses and intensivists adhered to established CAM-ICU and ICDSC protocols. Importantly, ICU nurses were not informed of this study’s details. Because this study was retrospective, the ICU nurses were not required to be informed about the research purpose. However, since information on whether VC was administered was recorded in the electronic medical records, it was not possible to blind this study completely. Secondary outcomes included ICU length of stay, postoperative hospitalization duration, cortisol level at ICU admission, maximum vasoactive-inotropic score (VIS), SOFA score on postoperative day 1, and cytokine levels. All data were retrieved retrospectively from electronic medical records.

### 2.5. Data Collection and Variables

Clinical data were retrospectively extracted from electronic health records. Patient characteristics included sex, age, body mass index, surgical method, smoking history, preoperative steroid use, preoperative benzodiazepine use, ASA classification (≥3), preoperative C-reactive protein (CRP), APACHE II score at ICU admission, and SOFA score. Preoperative comorbidities assessed included cognitive dysfunction, brain disease, lung disease, renal disease, low cardiac function (EF < 60%), liver dysfunction, diabetes, hypertension, dyslipidemia, heavy alcohol use, and obstructive sleep apnea syndrome. Intraoperative variables included operative time, anesthesia duration, CPB duration, cardiac arrest duration, selective cerebral perfusion duration, cardiac index (CI) after anesthesia induction, fluid balance, and doses of remifentanil and fentanyl. Postoperative variables included CI at ICU admission, leukocyte count, IL-6, maximum lactate levels from ICU admission to postoperative day 1, and the proportion of patients who were administered various sedatives. Of residual samples after routine blood gas analysis, 10 μL of plasma were stored frozen and used to comprehensively assess the levels of cytokines and chemokines. Data were collected using the same criteria and protocols for all patients. Missing data were excluded in principle; however, missing basic demographic data such as age and sex were supplemented from medical records.

### 2.6. Laboratory Measurements

#### 2.6.1. AFR/DMSO Method

When VC scavenges reactive oxygen species, it forms AFR, thus increasing AFR levels when reactive oxygen species are abundant. However, AFR is present only in very small amounts and is short-lived, making it undetectable by ESR instruments. AFR/DMSO, however, is entirely different from AFR and reflects the VC concentration itself.

When DMSO is added to plasma, electron spin resonance (ESR) detects a characteristic ascorbyl free radical (AFR) doublet (Figure 3). Six manganese (Mn^2+^) signals are also observed, which are configured in the ESR unit as internal standard markers. The AFR signal appears between Mn^2+^ signals 3 and 4. The relative intensity of AFR to Mn^2+^ signal 3 was calculated as plasma AFR/DMSO (Figure 2). Because AFR/DMSO values positively correlate with plasma VC levels measured by high-performance liquid chromatography (HPLC), VC levels can be determined in real time using AFR/DMSO measurements [16].

Plasma AFR/DMSO was measured at four time points: preoperatively (immediately after anesthesia induction), immediately after CPB withdrawal, at ICU admission, and on postoperative day 1. One milliliter of arterial blood was collected via a radial artery catheter. Immediately after routine blood gas analysis, plasma was separated from the remaining blood by centrifugation (4 °C, 3000× *g*, 10 min). Then, 50 μL of plasma and 100 μL of DMSO were mixed, transferred to a quartz cell, and inserted into the ESR resonator for measurement. AFR/DMSO levels were obtained within 2 min. The remaining plasma was stored at −80 °C for later cytokine analysis. ICU admission and postoperative day 1 samples from one NVC patient could not be measured due to hemolysis. Because the quartz cell used in prior studies [16] was replaced, AFR/DMSO levels in new healthy volunteers (n = 15) were 0.935 ± 0.052 [19]. Since AFR/DMSO is not a common marker, the normal values were cited as reference values but not used in the analysis.

#### 2.6.2. Cytokine Assays

Frozen plasma samples were thawed and cytokine levels measured using the Bio-Plex Pro Human GI 17-Plex Panel (M5000031YV; Bio-Rad, Hercules, CA, USA) according to the manufacturer’s instructions. Data were analyzed with Bio-Plex Manager software (version 6.0).

### 2.7. Sample Size Calculation

No prior studies have demonstrated that intravenous VC after cardiac surgery reduces POD incidence. Based on a previous report indicating that 46% of cardiac surgery patients developed POD [20], we assumed a 10% absolute reduction to be clinically meaningful. The required sample size was calculated as 757 patients at a significance level of 0.05 and a power of 0.8 in a two-tailed test. However, only 84 patients were available during the planned study period, resulting in a substantial lack of statistical power. Therefore, the results of this analysis should be considered exploratory.

### 2.8. Statistical Analysis

Continuous variables are expressed as medians (interquartile range [IQR]) and categorical variables as numbers (%). Intergroup comparisons of patient characteristics, preoperative comorbidities, intraoperative, and postoperative factors were performed using the Mann–Whitney U test, Fisher’s exact test, or chi-square test, as appropriate. The primary outcome, delirium incidence, was analyzed using the chi-square test, and odds ratios (ORs) with 95% confidence intervals (CIs) were calculated. To adjust for confounders (APACHE II score, preoperative benzodiazepine use, and VC administration), multivariate logistic regression analysis was conducted. Missing data were assumed to be missing at random and were supplemented by multiple imputation. Statistical analyses were performed using StatFlex ver. 7 (Artech, Osaka, Japan), with the significance level set at 5%.

Plasma AFR/DMSO levels are expressed as mean ± standard deviation. Two-way ANOVA followed by Tukey’s multiple comparison test was used to assess differences. These analyses were conducted using GraphPad Prism 8 for macOS (GraphPad Software, Boston, MA, USA). The significance level was set at 5%. Because of the small sample size, it was not possible to include all potential confounders and avoid overadjustment. Therefore, only the APACHE II score and preoperative BZ (Benzodiazepines) use reported in many studies as predictors of POD were included in the multivariate model.

## 3. Results

### 3.1. Participant Flow and Group Allocation

During the study period, 85 patients were enrolled. Because almost no remaining samples were available for one patient, the final analysis included 40 in the NVC group and 44 in the VC group (Figure 4).

### 3.2. Baseline Characteristics

Patient characteristics are shown in Table 1. Median age did not differ significantly between groups: 73.0 years (IQR 64.5–77.5) in the NVC group and 70.5 years (IQR 64.0–70.0) in the VC group. No significant differences were observed for other baseline characteristics. Similarly, no significant intergroup differences were found for preoperative comorbidities (Table 2) or intraoperative factors (Table 3). For postoperative factors (Table 4), no significant differences were found in inflammatory markers or medications used. However, maximum lactic acid levels after ICU admission were significantly higher in the NVC group (*p* = 0.02).

### 3.3. Plasma AFR/DMSO Levels (VC Status)

Figure 5 shows temporal changes in plasma AFR/DMSO values for both groups. In the NVC group, values were 0.630 ± 0.190 preoperatively (n = 40), 0.251 ± 0.095 immediately after CPB withdrawal (n = 40), 0.250 ± 0.084 at ICU admission (n = 39), and 0.320 ± 0.108 on postoperative day 1 (n = 39). In the VC group, values were 0.635 ± 0.220 preoperatively (n = 44), 0.255 ± 0.124 immediately after CPB withdrawal (n = 44), 0.268 ± 0.111 at ICU admission (n = 44), and 0.925 ± 0.316 on postoperative day 1 (n = 44). Multiple comparisons showed no significant difference between groups at baseline. In both groups, values immediately after CPB withdrawal and at ICU admission were significantly lower than preoperative values (*p* < 0.0001). On postoperative day 1, AFR/DMSO levels were markedly higher in the VC group than in the NVC group (0.925 ± 0.316 vs. 0.320 ± 0.108, *p* < 0.0001), supporting the role of VC replenishment in restoring antioxidant capacity.

### 3.4. Primary Outcome: Postoperative Delirium

The incidence of POD was significantly lower in the VC group than in the NVC group (11.4% [5/44] vs. 35.0% [14/40], *p* < 0.01) (Table 5). Multivariate logistic regression analysis with POD incidence as the dependent variable and APACHE II score, preoperative benzodiazepine use, and VC administration as explanatory variables demonstrated that VC administration was independently associated with a reduced risk of POD (adjusted OR 0.22, 95% CIs 0.07–0.69, *p* < 0.01) (Table 6).

### 3.5. Secondary Outcomes

No significant differences were found between groups for secondary outcomes (Table 5). After excluding cytokines with values below detection limits, comparisons of IL-6, IL-8, IL-10, G-CSF, IFN-γ, MCP-1, MIP-1β, and TNF-α revealed no significant differences between groups, except for IFN-γ and MCP-1 (Table 7). Although significant, changes in IFN-γ and MCP-1 remained within the normal range.

### 3.6. Safety Outcomes

Daily safety monitoring included morning blood biochemistry tests and chest X-rays, as well as blood gas analysis, blood glucose, electrolytes, lactate levels, and other blood tests every four hours. No serious adverse events occurred in either group. Retrospective review of medical records for gastrointestinal symptoms, administration-related reactions, and abnormal laboratory values revealed no minor adverse events attributable to VC.

## 4. Discussion

The study period was 6 months for both the NVC and VC groups. No significant differences were found between the groups in preoperative factors (Table 1), preoperative comorbidities (Table 2), or intraoperative factors (Table 3). Furthermore, no significant differences were observed in age, preoperative cognitive dysfunction, or types of drugs administered perioperatively [7], which are the known risk factors for POD. For postoperative factors, no significant differences were observed in leukocyte counts or IL-6 levels at ICU admission and in cytokine levels on postoperative day 1, except for maximum lactate levels, IFN-γ, and MCP-1 (Table 4 and Table 7). A recent study showed that a maximum lactate level ≥ 2.85 mmol/L on the first postoperative day significantly increases the risk of POD after cardiac surgery [21]. In our study, median lactate levels were 3.25 mmol/L in the NVC group and 4.55 mmol/L in the VC group, suggesting that both groups may have been at high risk of POD. IFN-γ and MCP-1 were significantly elevated in the VC group. Both IFN-γ and MCP-1 are central to inflammatory and neuroimmune pathways that could plausibly link oxidative stress modulation with POD risk. IFN-γ is a proinflammatory cytokine known to activate microglia and promote neuroinflammation [22]. MCP-1 (CCL2) recruits monocytes/macrophages and is associated with blood–brain barrier disruption and neuronal injury [23]. An increase in IFN-γ and MCP-1 in the VC group appears paradoxical if VC is hypothesized to attenuate neuroinflammation. However, it is possible that such increase is compensatory or transient, or biologically insignificant, because no significant differences were observed between the two groups in other inflammatory cytokines or chemokines. This suggests that the preventive effect of VC administration on POD was appropriately assessed. In this study, AFR/DMSO was measured as a surrogate marker for plasma VC concentration. Plasma AFR/DMSO levels were lower in patients undergoing CPB-assisted cardiac surgery than preoperatively and decreased further after CPB withdrawal and at ICU admission (Figure 5). Intravenous administration of 2 g VC immediately upon ICU admission significantly reduced POD incidence (Table 5), and multivariate logistic regression analysis confirmed that VC administration was independently associated with a lower risk of POD (Table 6). On postoperative day 1, AFR/DMSO levels remained low in the NVC group but were nearly threefold higher in the VC group before returning to baseline levels (Figure 5). Although we did not measure antioxidants other than AFR/DMSO, it is known that when blood is oxidized by air, VC decreases most rapidly, within one hour, whereas other antioxidants require four to five hours to be depleted [10]. Therefore, AFR/DMSO is the most sensitive marker of oxidative stress and antioxidant capacity. The recovery of AFR/DMSO on postoperative day 1 likely reflects restoration of antioxidant defense mechanisms and attenuation of oxidative stress, which is one of the key pathogenic mechanisms of POD. We therefore propose that a single intravenous bolus of 2 g VC at ICU admission not only suppresses oxidative stress but also restores the ROS–antioxidant balance.

The risk of POD is known to be high after cardiac surgery [1,20]. Its pathogenic mechanism is thought to involve CPB-induced systemic inflammatory response and oxidative stress, which cause vascular endothelial dysfunction and increase blood–brain barrier (BBB) permeability [24]. BBB disruption allows inflammatory cytokines and ROS to enter the brain, leading to neuronal injury and microglial activation, ultimately resulting in POD [7,25]. Both animal [26] and clinical studies [9,27] have demonstrated links between BBB failure and POD. Furthermore, when oxidative stress markers rise during cardiac surgery, POD incidence and neural injury increase, particularly in patients with marked BBB dysfunction [9]. These findings suggest that strategies targeting oxidative stress, inflammation, and BBB protection are crucial for POD prevention.

The VC used in this study functions as an antioxidant and plays a key role in oxidative stress regulation and neuroprotection. In animal studies, VC suppressed BBB failure induced by sustained compression of the primary somatosensory cortex [28] or by transient cerebral ischemia [29]. VC has also been reported to exert neuroprotective effects by suppressing inflammation. Specifically, LPS-induced microglial activation increases ROS production in the hippocampus of mice, contributing to neuronal injury and cognitive dysfunction. VC administration suppressed production of inflammatory cytokines by inhibiting microglial activation and significantly reduced malondialdehyde (MDA), an indicator of oxidative stress, thereby alleviating cognitive dysfunction [30]. Collectively, these findings suggest that VC’s ability to suppress oxidative stress and protect the BBB may contribute to POD prevention (Figure 6).

Several studies have reported the timing of oxidative stress and BBB failure after cardiac surgery with CPB. BBB failure biomarkers such as S-100B and MMP-9 peak at the completion of surgery [31], and BBB failure has been detected on MRI within 24 h after cardiac surgery [32]. Malondialdehyde (MDA) levels peak at the end of CPB and return to preoperative levels by postoperative day 1 [13]. In infants undergoing CPB for congenital heart disease, the lowest VC levels were observed at postoperative PICU admission [33]. Taken together, these findings suggest that oxidative stress after CPB peaks at the completion of surgery and then gradually decreases.

In the present study, we administered a single 2 g dose of VC upon ICU admission after surgery, at the time when oxidative stress and BBB failure are thought to peak, with the objective of restoring VC levels to normal. AFR/DMSO was measured to verify VC levels. Our results showed that VC administration at ICU admission may suppress POD.

However, no consistent conclusions have been reached regarding the inhibitory effect of VC on delirium in critically ill patients, such as those with sepsis. Achieving the benefits of VC depends on maintaining a balance between ROS and VC. Both timing and the dose are critical. With respect to timing, a prospective observational study on older patients undergoing cardiovascular surgery [34] did not show protective effects when patients received VC 500 mg before surgery, followed by 500 mg every 6 h for 2 days. Because VC was administered both before and long after surgery—instead of immediately after surgery when oxidative stress peaks—the ROS–VC balance may not have been maintained. With respect to dose, a sepsis study [35] reported that high-dose VC (50 mg/kg every 6 h) significantly worsened the composite outcome at day 28 (mortality plus organ failure) compared with placebo. This may have occurred because excessive VC depleted ROS below the level required for immune and cellular homeostasis, leading to oxidative stress due to antioxidant overload. Indeed, excessive antioxidant administration is known to cause free radical–induced DNA damage [36]. Moreover, a prospective study on [37] older cardiac surgery patients demonstrated that low postoperative, but not preoperative, VC levels were associated with POD. This suggests that monitoring VC levels is critical for identifying optimal timing and evaluating the effectiveness of supplementation, as well as predicting POD. The novelty of the present study lies not only in examining timing and dose but also in objectively assessing VC levels using AFR/DMSO.

## 5. Limitations of the Study

This study has limitations. First, power analysis indicated a required sample size of 757, but restricting the observation period to six months before and after the intervention resulted in only 84 cases, raising the risk of type II error. Because our ICU is the only high-intensity type ICU in the region, we have to accept not only postoperative patients but also critically ill patients with various conditions. Therefore, the required number of cases could not be reached within the 1-year study period. Second, the limited sample size prevented direct evaluation of the statistical association between AFR/DMSO changes and POD onset. Third, grouping based on calendar time (before vs. after April 2022) may have introduced temporal bias related to seasonal variation or practice changes. Fourth, residual confounding, including imbalances in lactate levels, cannot be fully excluded. Fifth, POD assessments were retrospectively extracted from medical records, leaving concerns about inter-rater reliability. Sixth, as a single-center, retrospective, before-and-after observational study, the evidence level is lower than that of prospective or RCTs. However, no significant differences were observed between groups in patient characteristics, comorbidities, or intraoperative and postoperative factors other than VC administration, suggesting these variables were unlikely to confound the findings. Despite these limitations, before-and-after studies can be conducted efficiently before RCTs, and this study provides meaningful comparative data on the potential inhibitory effect of VC on POD. Future multicenter RCTs are required to confirm reproducibility and establish an optimal VC administration strategy.

## 6. Conclusions

To maximize the protective effect of VC against POD, it is essential to determine the appropriate timing, dose, and duration of administration. In this study, intravenous administration of 2 g VC at ICU admission after CPB-assisted cardiac surgery restored VC levels to normal by postoperative day 1 and was significantly associated with a reduced risk of POD. Unlike in sepsis, oxidative stress after CPB appears to peak at surgery completion and decline rapidly thereafter. Therefore, a therapeutic strategy that administers VC at the peak of oxidative stress may be beneficial. Verification of these findings in RCTs is warranted in the future.

## Figures and Tables

**Figure 1 jcm-15-00135-f001:**
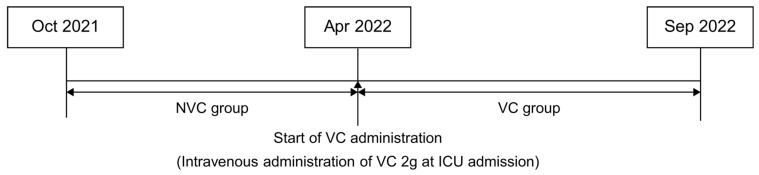
Study flow. Beginning in April 2022, intravenous administration of 2 g of vitamin C (VC) upon postoperative intensive care unit admission was initiated for patients undergoing cardiac surgery with cardiopulmonary bypass. NVC, non-VC group; VC, vitamin C.

**Figure 2 jcm-15-00135-f002:**
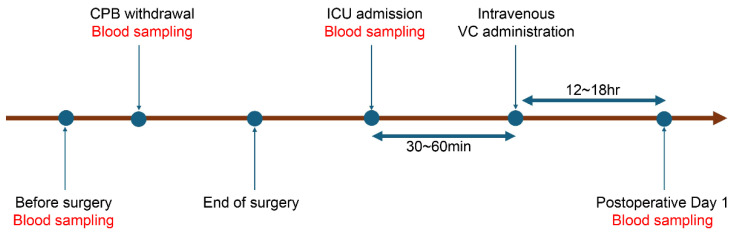
Time course of blood sampling and intravenous vitamin C (VC) administration.

**Figure 3 jcm-15-00135-f003:**
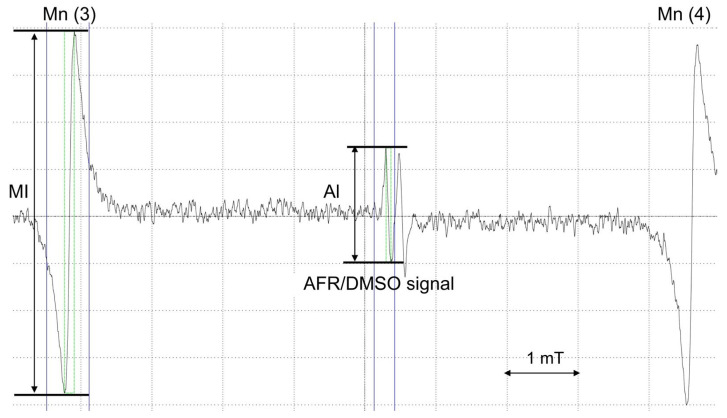
Plasma AFR/DMSO detected by the ESR spectrometer. AI/MI is defined as the ratio of AFR/DMSO to manganese intensity. ESR, electron spin resonance; AFR/DMSO, ascorbyl free radical detected by adding dimethyl sulfoxide; AI, AFR/DMSO intensity; MI, Manganese intensity.

**Figure 4 jcm-15-00135-f004:**
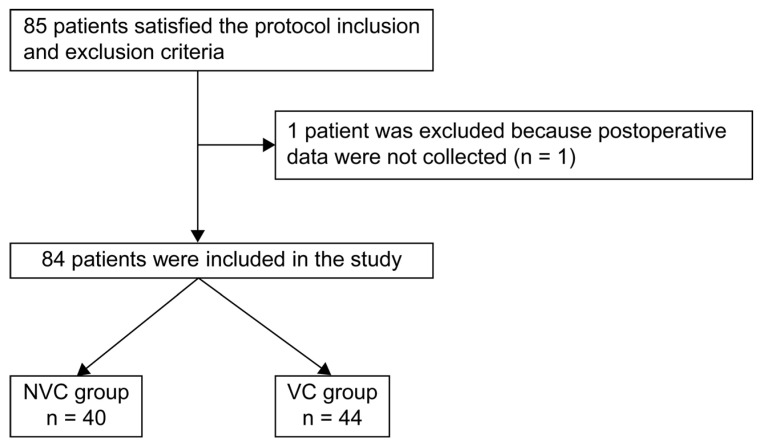
Flowchart of the patient inclusion. NVC, non-vitamin C; VC, vitamin C.

**Figure 5 jcm-15-00135-f005:**
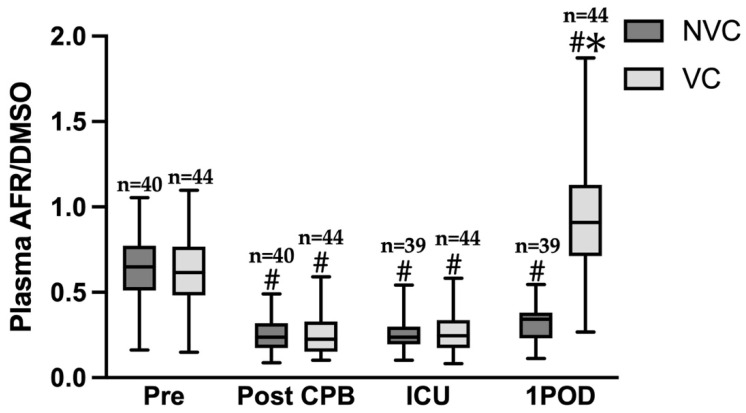
Plasma AFR/DMSO in the NVC or VC group after cardiac surgery. Tukey’s multiple comparisons test. Results are presented as means ± standard deviation. Sharp represents statistical significance when compared to Pre (*p* < 0.0001). Asterisk represents statistical significance when compared to NVC and VC (*p* < 0.0001). Pre, immediately after induction of anesthesia; Post CPB, just after the end of cardiopulmonary bypass; ICU, immediately after admission to the ICU; 1POD, postoperative day 1; NVC, non-vitamin C; VC, vitamin C.

**Figure 6 jcm-15-00135-f006:**
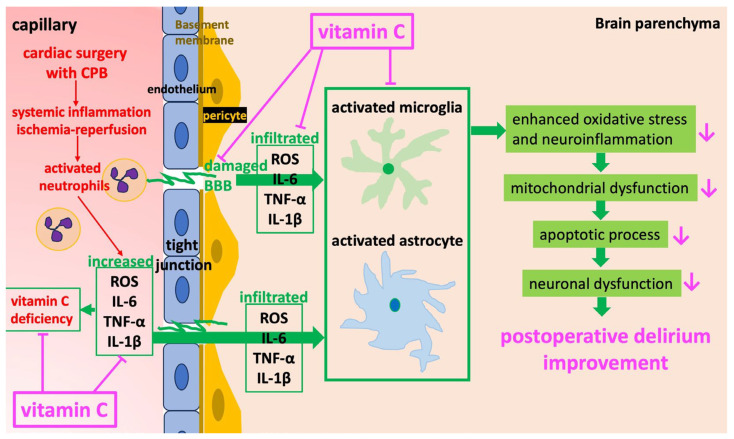
Effects of vitamin C on BBB disruptions and activation of microglia and astrocytes after cardiac surgery. Following cardiac surgery under CPB, neutrophils are activated due to systemic inflammatory response and ischemia–reperfusion. Activated neutrophils adhere to tight junctions of the BBB, as well as to pericytes and the basement membrane, releasing large amounts of inflammatory cytokines and reactive oxygen species, thereby causing BBB disruption. Through a disrupted BBB, excessive ROS and cytokines infiltrate the brain parenchyma and activate microglia and astroglia. Activated microglia and astrocytes further induce inflammatory mediators and ROS, which cause mitochondrial dysfunction and activate the apoptosis signaling pathway. These processes ultimately lead to neuronal damage. BBB, blood–brain barrier; CBP, cardiopulmonary bypass; ROS, reactive oxygen species. T-shaped lines indicate inhibition.

**Table 1 jcm-15-00135-t001:** Patient characteristics.

	NVC Group (n = 40)	VC Group (n = 44)	*p* Value
Sex (M/F)	26/14	28/16	n.s.
Age (years)	73.0 (64.5–77.5)	70.5 (64.0–75.0)	n.s.
BMI (kg/m^2^)	23.1 (21.8–25.1)	20.8 (21.8–25.1)	n.s.
Surgical method (%)			n.s.
Ascending aorta replacement	3 (7.5)	1 (2.3)	
Arch replacement	6 (15.0)	8 (18.2)	
Root replacement	2 (5.0)	1 (2.3)	
Valve replacement	26 (65.0)	30 (68.2)	
Coronary artery replacement	1 (2.5)	0 (0.0)	
Descending aorta replacement	2 (5.0)	0 (0.0)	
Other open-heart surgery	0 (0.0)	4 (9.1)	
Smoking history (%)	24 (60.0)	26 (59.1)	n.s.
Preoperative steroid use (%)	2 (4.5)	1 (2.5)	n.s.
Preoperative BZ use (%)	1 (2.5)	2 (4.5)	n.s.
ASA classification ≥ 3	28 (70.0)	31 (70.5)	n.s.
Preoperative CRP (mg/dL)	0.09 (0.04–0.29)	0.08 (0.04–0.14)	n.s.
APACHE II on ICU admission	10.0 (7.5–12.5)	11.0 (9.0–13.5)	n.s.
SOFA on ICU admission	5.0 (4.0–7.0)	4.0 (2.0–6.0)	n.s.

Data are presented as median (interquartile range) or number (n) of patients (%). n.s., not significant; BMI, body mass index; BZ, benzodiazepines; ASA, American Society of Anesthesiologists; CRP, C-Reactive Protein; ICU, intensive care unit; APACHE, Acute Physiologic Assessment and Chronic Health Evaluation; SOFA, Sequential Organ Failure Assessment; NVC, non-vitamin C; VC, vitamin C.

**Table 2 jcm-15-00135-t002:** Preoperative comorbidities.

	NVC Group (n = 40)	VC Group (n = 44)	*p*-Value
Cognitive dysfunction (%)	1 (2.5)	1 (2.3)	n.s.
Brain disease (%)			n.s.
Cerebral infarction	3 (7.5)	7 (15.9)	
Subarachnoid hemorrhage	2 (5.0)	1 (2.3)	
Lung disease (%)			n.s.
Obstructive respiratory event	8 (20.0)	9 (20.5)	
Restrictive ventilatory impairment	6 (15.0)	3 (6.8)	
Combined ventilatory impairment	5 (12.5)	3 (6.8)	
Unknown	1 (2.5)	1 (2.3)	
Renal disease (%)			n.s.
Chronic kidney disease	18 (45.0)	21 (47.7)	
Dialysis	1 (2.5)	1 (2.3)	
Low cardiac function (%)	13 (32.5)	14 (31.8)	n.s.
Liver dysfunction (%)	9 (22.5)	4 (9.1)	n.s.
Diabetes (%)	13 (32.5)	8 (18.2)	n.s.
Hypertension (%)	33 (82.5)	33 (75.0)	n.s.
Dyslipidemia (%)	18 (45.0)	18 (40.9)	n.s.
Heavy alcohol use (%)	1 (2.5)	0 (0.0)	n.s.
Obstructive sleep apnea syndrome (%)	1 (2.5)	2 (4.5)	n.s.

Data are presented as number (n) of patients (%). n.s., not significant. NVC, non-vitamin C; VC, vitamin C.

**Table 3 jcm-15-00135-t003:** Intraoperative factors.

	NVC Group (n = 40)	VC Group (n = 44)	*p*-Value
Operative time (mins)	409.5 (309.0–484.0)	374.0 (313.0–439.5)	n.s.
Anesthesia duration (mins)	537.0 (433.0–614.0)	522.0 (449.0–573.0)	n.s.
Duration of CPB (mins)	201.0 (162.0–271.0)	212.0 (171.5–251.0)	n.s.
Duration of cardiac arrest (mins)	155.5 (124.5–192.5)	151.5 (115.5–186.0)	n.s.
Duration of selective cerebral perfusion (mins)	0.0 (0.0–0.0)	0.0 (0.0–0.0)	n.s.
Post-anesthesia induction CI (L/min/m^2^)	2.19 (1.62–2.66)	2.15 (1.78–2.52)	n.s.
Infusion balance (mL)	1780 (910–2445)	1240 (330–1920)	n.s.
Remifentanil (mg/kg)	0.085 (0.065–0.125)	0.100 (0.075–0.110)	n.s.
Fentanyl (µg/kg)	9.2 (7.8–10.7)	10.1 (8.0–13.2)	n.s.

Data presented as median (interquartile range). n.s., not significant; CI, cardiac index.

**Table 4 jcm-15-00135-t004:** Postoperative factors.

	NVC Group (n = 40)	VC Group (n = 44)	*p*-Value
CI (L/min/m^2^) after ICU admission	2.40 (1.90–3.17)	2.57 (2.21–3.75)	n.s.
WBC counts (/μL)	9485 (7475–10,975)	10,020 (7640–12,460)	n.s.
IL-6 (pg/mL)	374.5 (267.0–546.5)	332.0 (171.0–498.0)	n.s.
Maximum lactate (mmol/L) after ICU admission	3.25 (2.30–4.65)	4.55 (3.15–5.15)	0.02
Sedatives used in ICU			
Propofol (%)	39 (97.5)	44 (100.0)	n.s.
Midazolam (%)	4 (10.0)	4 (9.1)	n.s.
Dexmedetomidine (%)	32 (80.0)	35 (79.5)	n.s.

Data presented as median (interquartile range) or number (n) of patients (%). n.s., not significant; ICU, intensive care unit; CI, cardiac index; NVC, non-vitamin C; VC, vitamin C; WBC, white blood cell; IL-6, interleukin-6.

**Table 5 jcm-15-00135-t005:** Primary and secondary outcomes.

	NVC Group (n = 40)	VC Group (n = 44)	*p*-Value
Primary outcome			
POD incidence rate (%)	14 (35.0)	5 (11.4)	<0.01
Secondary outcome			
SOFA score after ICU admission	6.5 (5.0–8.0)	7.0 (3.0–7.5)	n.s.
Cortisol level (µg/dL)	20.7 (16.5–28.2)	21.6 (15.6–29.3)	n.s.
Maximum VIS	8.58 (4.30–11.68)	8.78 (6.75–13.6)	n.s.
ICU length of stay (days)	3.0 (3.0–5.0)	3.0 (3.0–5.0)	n.s.
Postoperative duration of hospitalization (days)	24.0 (22.0–36.0)	23.0 (20.5–24.5)	n.s.

Data presented as median (interquartile range) or number (n) of patients (%). n.s., not significant; SOFA, Sequential Organ Failure Assessment; VIS, vasoactive inotropic score; ICU, intensive care unit; VIS; Vasoactive-Inotropic Score dopamine dose [µg/kg/min] + dobutamine [µg/kg/min] + 100 × norepinephrine dose [µg/kg/min] + 10 × milrinone dose [µg/kg/min] + 25 × Olprinone dose [µg/kg/min] + 10,000 × vasopressin [units/kg/min].

**Table 6 jcm-15-00135-t006:** Multivariate logistic regression analysis.

Variable	Odds Ratio (95% CIs)	*p*-Value
APACHE II	1.08 (0.94–1.25)	1.05
Preoperative BZ use	1.75 (0.09–32.3)	0.38
VC administration	0.22 (0.07–0.69)	<0.01

Data presented as median (interquartile range) or number (n) of patients (%). CIs, confidence intervals; APACHE, Acute Physiologic Assessment and Chronic Health Evaluation; BZ, benzodiazepines; VC, vitamin C.

**Table 7 jcm-15-00135-t007:** Secondary outcome (cytokines).

	NVC Group (n = 40)	VC Group (n = 44)	*p*-Value
IL-6 (pg/mL)	87.8 (59.8–127.9)	77.3 (53.8–123.9)	n.s.
IL-8 (pg/mL)	58.2 (38.6–78.8)	49.4 (30.9–76.3)	n.s.
IL-10 (pg/mL)	14.1 (9.5–27.9)	11.2 (5.9–22.1)	n.s.
G-CSF (pg/mL)	103.7 (69.7–152.5)	79.1 (50.2–117.8)	n.s.
IFN-γ (pg/mL)	0.97 (0.38–3.31)	4.23 (2.35–11.1)	<0.01
MCP-1 (pg/mL)	87.6 (56.3–161.1)	129.8 (92.3–233.3)	<0.01
MIP-1b (pg/mL)	26.8 (19.0–40.8)	29.6 (18.0–52.7)	n.s.
TNF-α (pg/mL)	24.2 (15.6–34.2)	24.7 (17.3–35.7)	n.s.

Data presented as median (interquartile range). n.s., not significant; IL-6, interleukin-6; IL-8, interleukin-8; IL-10; interleukin-10; G-CSF; granulocyte-colony stimulating factor; INF-γ, interferon gamma MCP-1, monocyte chemoattractant protein-1; MIP-1b, Macrophage inflammatory protein-1; TNF-α, Tumor Necrosis Factor-α.

## Data Availability

The data presented in this study are available on request from the corresponding author. The data are not publicly available due to privacy or ethical restrictions.

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
