# Peer review of "Association of Vitamin C Administration with Postoperative Delirium After Cardiac Surgery with Cardiopulmonary Bypass: A Single-Center Retrospective Exploratory Cohort Studyâ€"

_jcm, 2025, doi:10.3390/jcm15010135_

Round 1
Reviewer 1 Report
Comments and Suggestions for Authors
Review Vitamin C:
This manuscript compares two patient groups in regards to the POD incidence in patients that
underwent cardiac surgery during their ICU stay. The reduction of POD in this sensitive patient group is
an important goal. In the retrospective study one patient group received 2g Vitamin C after surgery and
the other one did not. In both groups POD was assessed on the first post-operative day. POD occurred
in less cases in the patient group that received Vitamin C after surgery.
1. In the abstract and in the text the authors state that lactate level was higher in the NVC group
(control group). Table 4 displayed a significant higher lactate value in the VC group. This
discrepancy should be reviewed and corrected. And this fact should be discussed considering
that there has been at least one study (An, R.; Wu, X.; Bie, D.; Ding, J.; Li, Y.; Jia, Y.; Yuan, S.; Yan,
F. Association between the highest lactate level on the first postoperative Day and
postoperative delirium in cardiac surgery patients. CNS Neurosci Ther 2025) where POD was
associated with a higher post-operative lactate level.
2. It would be interesting to know why Vitamin C administration on ICU admission was
implemented. Was it because of the study or due to some other reasons? It would be
interesting to know why the measurement of ascorbyl free radical/dimethyl sulfoxide
(AFR/DMSO) was not done in all patients, but only in a small group.
3. When it was implemented for all patients it should be explained why they were not able to
include more patients in each study group. Another question is – as this is a retrospective study
– why the authors did not just compare a higher number of patients during a longer study
period. This would support the results more.
4. The small number of patients is in my opinion a major shortcoming of the study. The authors
stated that “the sample size was calculated as 757 patients at a significance level of 0.05 and
power of 0.8 in a two-tailed test. However, only 84 patients were available during the planned
study period, resulting in a substantial lack of statistical power. Therefore, the results of this
analysis should be considered exploratory.”
I think it would be better to state the exploratory character of the study in the heading and in
the abstract.
5. The authors stated that the primary outcome was POD during ICU admission, but the exact
timeframe is missing. Was POD assessment done for one day or more and when did it start?
The manuscript should provide a more clear picture of the POD assessment period as this
assessment generates the primary outcome.
6. The authors pointed out that the ICU nurses were not informed about the study details. But as
this study is considered a retrospective study this seems not necessary. It would be helpful, if
the authors could explain how knowledge of the study might have influenced the nurses. As
this was not a blinded study the nurses might have know that the patients received Vitamin C
through the patient record where the administration was documented.
7. The section 2.5 reads a bit like a mixture of retrospective and prospective data assessment. It
seems that at least the cytokines, the lactate, and the sedative level were measured exclusively
in study patients. This should be explained in more details in the manuscript.
8. In line 192 and 193 the authors state that “ AFR/DMSO levels in new healthy volunteers (n =
15) were measured”. This contradicts the statement that the control group and the
intervention group comprised of cardio-surgical patients and were comparable and seems to
refer to another study.
9. The authors write that “because of the small sample size, it was not possible to include all
potential confounders in the multivariate model”. I think it should be discussed whether a
multivariate analysis could be used in this patient group and it would be nice to get an
explanation why the authors choose APACHE II score and pre-operative benzodiazepine use as
variables for the multivariate regression analysis.
10. It would be interesting for the reader to discuss the fact that IFN-γ and MCP-1 level differed
significantly in both study groups. Has this occurred in other studies?
11. I would like to mention that the term „elderly patients“ should no longer be used. It is
recommended to use the term “older patients” instead.

Reviewer 2 Report
Comments and Suggestions for Authors
The information from lines 17- 19 ( We used electron spin resonance (ESR) to monitor vitamin C (VC) in real time and conducted a before-and-after study of intravenous VC (2 g) administered upon admission to the intensive care unit (ICU) after surgery) in the abstract should be in the methods section, not in the objectives. The aim should also be better formulated in the objectives subsection of the abstract.
However, the limitations described by the authors (First, power analysis indicated a required sample 391 size of 757, but restricting the observation period to six months before and after the intervention resulted in only 84 cases, raising the risk of type II error. Second, the limited sample size prevented direct evaluation of the statistical association between AFR/DMSO 394 changes and POD onset. Third, grouping based on calendar time (before vs. after April 2022) may have introduced temporal bias related to seasonal variation or practice changes. Fourth, residual confounding, including imbalances in lactate levels, cannot be fully excluded. Fifth, POD assessments were retrospectively extracted from medical records, leaving concerns about inter-rater reliability) represent indeed serious factors that could have influenced the results of the study decisively.
Reviewer 3 Report
Comments and Suggestions for Authors
The manuscript by Yoshihide Kuribayashi and colleagues describes a study of the vitamin C effect on postoperative delirium after cardiac surgery. Despite vitamin C is a relatively well studied molecule, search of its potential effects in various models and useful applications is still relevant. Vitamins in general and their effects and mechanisms of action remain an underestimated area of research. This study can provide a valuable strategy to reduce delirium in patients also giving data for the studies of delirium molecular mechanisms. However, the work should be described more precisely, and corrected. Here is a list for a point-by-point revision.
- More methodological details should be included. For example, a scheme illustrating "VC (2 g) administered upon admission to the intensive care unit (ICU) after surgery" should be added to Fig. 1, because description of details, time intervals especially, is missing.
- Similarly, other Methods should be rewritten to make their protocol and the measured values clearer. For example, the chapter on "AFR/DMSO method" includes a part of the method in the end, while an incomplete description is provided in the beginning. Please state, what this parameter indicates of, because as I understand, AFR depends both on the concentration of plasma VC and also on the redox system state.
- Mistakes should be corrected, and the text must be checked for more ones. Particularly, the manuscript states, that "lactate levels were higher in the NVC group" (line 28), but the levels of lactate is 40% higher in VC group (table 3). Moreover, the abstract must include a more precise description, including exact values. In this case the 40% change should be mentioned.
- The exact values can be removed to a supplementary table (lines 255-260), because they currently shade the important differences (lines 262-263), shown in the figure. Why such a decrease in AFR/DMSO occur? Is it caused by a reduction of VC concentration or reduction in redox system capacity?
- Lines 309-329 are no a discussion, and should be transferred to results.
- Line 391 should be entitled as a subchapter "limitations of the study".
- Fig. 4 should be presented as a box- or violin-plot.
- Discussion of vitamin C molecular mechanism of action should be added, and a potential scheme could be added. The latter should nit be extremely speculative, but rather be based on references. However, authors' interpretations and point of view can take place.
Round 2
Reviewer 1 Report
Comments and Suggestions for Authors
Dear Sirs,
I think that the revision has addressed the reviewer comments appropriately. There are only two minor points that I would like to hint at:
- In the abstract higher lactate levels are still assigned to the NVC group. This should be corrected to VC group.
- In line 246 BZ is mentioned for the first time without explanation of this abbreviation. Even though it is clear this abbreviation should be explained at first use.
Author Response
- In the abstract higher lactate levels are still assigned to the NVC group. This should be corrected to VC group
[Response]
We thank you for the careful review. As you pointed out, in the “Manuscript Information Overview”(https://susy.mdpi.com/user/manuscripts/review_info/9eca79b6cb308d20c1e13421dc420f99#author_services_section), higher lactate levels are still assigned to the NVC group, but in the Manuscript File (manuscript.docx) and PDF File (manuscript.pdf), this has been corrected to VC group. Could you please check this?
-
In line 246 BZ is mentioned for the first time without explanation of this abbreviation. Even though it is clear this abbreviation should be explained at first use.
[Response]
We thank you for the careful review. We added (Benzodiazepines) to BZ in line 246.
Reviewer 2 Report
Comments and Suggestions for Authors
The authors have properly addressed my concerns regarding the manuscript.
Author Response
- The authors have properly addressed my concerns regarding the manuscript.
[Response]
We appreciate your insightful review.
Reviewer 3 Report
Comments and Suggestions for Authors
The revised manuscript jcm-3979920 has progressed substantially. Here are a few minor comments remaining to be solved.
3. Speaking of lactate, please, add the value (40%) to the abstract.
5. If the other reviewers ask for an extension of this paragraph, I can't but agree. Otherwise, The results and discussion could be divided in a more precise manner.
7. Add also the "n=..." values to the figure (now Figure 5).
8. Good to see such a nice figure being added.
Author Response
- Speaking of lactate, please, add the value (40%) to the abstract.
[Response]
We thank you for your useful suggestions. We added the value (40%) to the abstract.
-
If the other reviewers ask for an extension of this paragraph, I can't but agree. Otherwise, The results and discussion could be divided in a more precise manner.
[Response]
We sincerely appreciate your understanding of our revisions.
- Add also the "n=..." values to the figure (now Figure 5).
[Response]
We appreciate your helpful suggestions. We added "n=..." values to Figure 5.
- Good to see such a nice figure being added.
[Response]
We appreciate your excellent comment.